# Short-Term Maize Rotation Suppresses Verticillium Wilt and Restructures Soil Microbiomes in Xinjiang Cotton Fields

**DOI:** 10.3390/microorganisms13091968

**Published:** 2025-08-22

**Authors:** Faisal Hayat Khan, Zhanjiang Tie, Xueqin Zhang, Yanjun Ma, Yu Yu, Sifeng Zhao, Xuekun Zhang, Hui Xi

**Affiliations:** 1Key Laboratory of Oasis Agricultural Pest Management and Plant Protection Resources Utilization, College of Agriculture, Shihezi University, Shihezi 832003, China; fhayat666@gmail.com (F.H.K.); t2424395514@163.com (Z.T.); zhang123451256@163.com (X.Z.); 15630077211@163.com (Y.M.); zhsf_agr@shzu.edu.cn (S.Z.); 2Xinjiang Academy of Agricultural Reclamation Sciences, Shihezi 832003, China; xjyuyu021@sohu.com; 3Workstation of Academician Zhang Xianlong, College of Agriculture, Shihezi University, Shihezi 832003, China

**Keywords:** crop rotation, cotton, Verticillium wilt, soil microbiome

## Abstract

Verticillium wilt, a prevalent soil-borne disease, poses a significant challenge to cotton production in Xinjiang, China. Continuous cotton monoculture has increased disease incidence and affected soil microbial diversity in Xinjiang, while crop rotation is recognized as an effective strategy for soil pathogen control. This study investigates how a one-year maize rotation affects Verticillium wilt incidence and soil microbiome composition in cotton fields across northern and southern Xinjiang. The results demonstrated that short-term rotation significantly reduced Verticillium wilt occurrence in both northern and southern Xinjiang. Using high-throughput sequencing of fungal ITS and bacterial 16S rRNA regions, microbial community analysis revealed minimal changes in alpha-diversity but significant structural reorganization between continuous cropping (CC) and rotation (CR) systems, particularly in fungal and bacterial genera composition, with distinct spatial patterns between northern and southern fields. Crop rotation promoted beneficial taxa such as *Sphingomonas* and *Pseudogymnoascus*, while reducing the abundance of pathogens such as *Verticillium dahliae*. LEfSe study suggested *Tepidisphaerales* and *Lasiosphaeriaceae* as biomarkers for CR systems, whereas *Hypocreales* and *Blastocatellia* dominated in CC soils. Co-occurrence network analysis revealed more bacterial connectivity and modularity under CR, suggesting better microbial interactions and ecological resilience. The increased structural complexity of bacterial networks under CR indicates their greater contribution to soil health maintenance and ecosystem resilience. Our findings demonstrate that short-term crop rotation not only effectively reduces Verticillium wilt incidence but also restructures soil microbial communities, providing an actionable strategy for sustainable cotton cultivation in Xinjiang.

## 1. Introduction

As the world’s leading cotton producer (*Gossypium* spp.), China occupies a pivotal position in the global cotton industry, with its output accounting for approximately 20% of the world’s total production. Domestically, the Xinjiang Uygur Autonomous Region is the core production area, contributing 90% of the national cotton output [1]. To maintain this national strategic target (annual output exceeding 5 million tons), continuous monoculture has become the main planting pattern in Xinjiang’s cotton fields. However, this intensive, long-term, continuous cropping system has a profound and cumulative impact on the soil ecosystem. With the increase in the number of continuous cropping years, the incidence of soil-borne pathogens gradually increases, causing a serious reduction in crop yields [2,3]. Over time, this causes a disruption in the natural balance of the microorganisms, which in turn causes an increase in the incidence of Verticillium wilt [4,5]. This is mainly caused by the emergence of newly developed virulent strains of *V. dahliae* [6]. As a soil-borne vascular pathogen, *V. dahliae* causes significant yield losses, especially in cotton, where most varieties have limited resistance, thus exerting a severe impact on agricultural production [7]. Given this persistent phytopathological challenge, crop rotation offers a cost-effective strategy for disease management by disrupting the ecological link between plant hosts and soil-borne pathogens [8]. Consequently, in cotton fields where Verticillium wilt outbreaks are severe, farmers often resort to emergency one-year rotation measures, choosing to rotate with Poaceae crops such as maize and wheat. Maize is an important feed crop with significant value; its economic importance is increasing with the growth of consumer demand and its use in biofuel production [9]. Existing studies have shown that the maize–cotton rotation pattern is capable of significantly enriching the beneficial microbial community in the cotton rhizosphere, and in turn enhancing cotton’s resistance to *V. dahliae* [10]. Due to the high economic value of cotton, farmers tend to shorten rotation periods in order to quickly resume cotton cultivation. While this practice may offer short-term economic gains, it invisibly undermines the long-term effectiveness of crop rotation as a sustainable strategy to control soil-borne diseases. Although short-term rotation has been widely applied in actual farming production systems, there still exists a critical gap in the mechanistic research regarding how the soil microbiome regulates the population size of *V. dahliae* and influences plant health status, and a clear understanding of this has not yet been formed. Soil microorganisms play an important role in decomposing soil organic matter, nutrient transformation, and soil-borne diseases [11].

Following infection, plants might recruit microorganisms that enhance disease resistance and stimulate growth, thereby influencing the root microbiome composition [12]. For instance, specific rhizosphere *Flavobacterium* spp. strains augment wilt resistance in tomato [13]. These protective activities, which are mediated by the soil microbiome, are responsible for regulating essential processes such as the decomposition of organic matter, the cycling of nutrients, and the control of diseases [11]. Crucially, root-associated microbial communities dynamically respond to soil–plant system changes, creating feedback loops that critically influence plant health outcomes [14]. However, under continuous cropping systems, microbial dysbiosis develops and directly contributes to cultivation obstacles through functional impairment. Specifically, monoculture progressively alters fungal–bacterial diversity, ecological network stability, and functional profiles across growing seasons [15,16]. This restructuring facilitates pathogenic fungal enrichment, which directly inhibits crop growth, constituting the core mechanism of these obstacles; accumulation of these harmful microorganisms under long-term monoculture conditions disrupts the balance of the soil microbiome, suppresses beneficial microbes, and fosters a disease-conducive environment [16,17]; pathogenic dominance is a core mechanism of biological disorders associated with continuous cropping, ultimately compromising plant health and reducing yield potential. Consequently, an emerging consensus identifies systemic microbiome restructuring as the primary driver of monoculture-induced cultivation constraints [15,16,18]. The imbalance of soil microbial community structure can be the main reason for the obstacle to continuous cropping. Continuous cropping had significant effects on the diversity, function, and co-occurrence networks of soil fungi and bacteria with inter-annual changes [16]. Upon the increase in continuous cropping years, the number of pathogens in soil and inter-rhizosphere increased, while the number of beneficial bacteria decreased [19].

Crop rotation fundamentally alters the factors influencing soil microbial community structure and spatial distribution by introducing temporal and biological diversity into agroecosystems. This practice mitigates pathogen pressure through a suite of complementary mechanisms and promotes the proliferation of antagonistic microorganisms and accumulation of inhibitory compounds, which can suppress pathogenic taxa through competition, antibiosis, or induced systemic resistance in host plants [20,21,22]. Despite these benefits, integrated strategies involving rotation with non-hosts are essential for Verticillium wilt control [23], yet research on the effects of short-term rotations on soil fungal and bacterial communities remains limited. Therefore, we conducted this study to (i) compare Verticillium wilt incidence in rotated versus continuous cotton systems; (ii) analyze structural shifts in soil microbial communities across northern and southern Xinjiang under one-year maize rotation; and (iii) elucidate rotation-induced interactions among cotton, *V. dahliae*, and the soil microbiome. Collectively, this study revealed the impact of cotton-maize on the northern and southern Xinjiang for the first time and provided a theoretical basis for the sustainable production of cotton in Xinjiang.

## 2. Materials and Methods

### 2.1. Field Site Selection and Verticillium Wilt Assessment

Field investigations were carried out during the peak period of Verticillium wilt incidence (August 2023), across major representative cotton-growing regions in northern and southern Xinjiang (Appendix A). Paired agricultural systems for comparison, including continuous cotton monoculture (CC) and maize–cotton rotation (CR), were selected from adjacent plots situated (<2 km separation) sharing equivalent edaphic and climatic conditions. In total, 26 fields comprising (13 pairs) CC and CR underwent systematic sampling and agronomic evaluation (Appendix A). The prevalence of Verticillium wilt within each field was determined using a standardized five-point diagonal sampling method, while disease severity was quantitatively assessed according to well-established five-level rating criteria widely used in plant pathology research [24]. This method involves visually assessing the extent of plant disease symptoms, usually focusing on leaves, and assigning a score from 1 to 5 based on the observed severity [25,26], as follows: (0 = no symptoms, 1 = 20~25% leaves showing chlorosis, 2 = 40~60% symptomatic leaves, 3 = 60~80% symptomatic leaves or partial plant wilting, 4 = 80~100% symptomatic leaves, severe wilting, or plant death).

### 2.2. Soil Sampling and Analysis

Soil cores (5 cm diameter × 20 cm depth) were systematically collected from each field using a standardized five-point sampling grid. Fifteen bulk soil cores per field were homogenized to form three composite samples (each comprising five randomly selected cores). This yielded five representative composites per field (three composites from the grid points). In order to collect the rhizosphere soil (1–2 mm thick soil layer around the roots after severe shaking), the roots were transferred to a 50 mL centrifuge tube containing 15 mL 1 × phosphate buffer solution (PBS), rotated for 5 min, and then taken out. Next, the test tube was centrifuged at 4000 rpm and 4 °C for 10 min, and the supernatant was discarded. Then, the sample was centrifuged at 8000 rpm for 5 min, the supernatant was discarded again, and the remaining part was regarded as rhizosphere soil [27]. All samples were immediately flash-frozen in dry ice during transport and maintained at −80 °C until DNA extraction. The experimental design generated 130 bulk soil samples (2 cropping systems [CC/CR] × 13 geographic sites × 5 biological replicates) for subsequent molecular analyses.

### 2.3. DNA Extraction and High-Throughput Sequencing

Total genomic DNA was extracted from 130 soil samples using the DNeasy PowerSoil Kit (QIAGEN, Hilden, Germany) following the manufacturer’s instructions. The extracted DNA was subjected to 0.8% agarose gel electrophoresis to determine the molecular size, and Nanodrop was used for DNA quantification. For fungal community characterization, the ITS1 region was amplified with primers ITS1F (5′-GGAAGTAAAAGTCGTAACAAGG-3′) and ITS2 (5′-GCTGCGTTCTTCATCGATGC-3′) [28]. Bacterial community analysis targeted the 16S rRNA V3-V4 hypervariable region using primers 341F (5′-ACTCCTACGGGAGGCAGCA-3′) and 805R (5′-GGACTACHVGGGTWTCTAAT-3′) [29]. PCR amplification was conducted under the following conditions: initial denaturation at 95 °C for 2 min; 30 cycles of 95 °C for 30 s, 55 °C for 30 s (annealing), and 72 °C for 30 s (extension); with final elongation at 72 °C for 5 min. PCR amplicons were purified with Vazyme VAHTSTM DNA Clean Beads (Vazyme, Nanjing, China) and quantified using the Quant-iT PicoGreen dsDNA Assay Kit (Invitrogen, Carlsbad, CA, USA). After the individual quantification step, amplicons were pooled in equal amounts, and paired-end sequencing (2 × 300 bp) was performed using the Illlumina MiSeq platform at Personal Biotechnology Co., Ltd. (Nanjing, China).

### 2.4. Bioinformatics and Statistical Analyses

Sequence processing was conducted in QIIME 2 using DADA2 (v1.20.0) for quality control and chimera removal [30]. Taxonomic assignment of 16S rRNA and ITS sequence variants (ASVs) employed the Greengenes (v13_8) and UNITE (v8.3) databases, respectively. The specific steps included demultiplexing raw sequences using the demux plugin and trimming primers with cutadapt; for quality control, forward reads were truncated to 240 bp and reverse reads to 180 bp (with a quality score threshold of Q ≥ 20), and sequences containing ambiguous bases (N) or with lengths < 150 bp were discarded; denoising was performed with maxEE = c(2, 3), and chimeras were removed using the removeBimeraDenovo function in “consensus” mode. Amplicon Sequence Variants (ASVs) were obtained by clustering sequences with 100% similarity, and rare ASVs with a total abundance < 0.001% were excluded; non-singleton ASVs of 16S rRNA gene data were aligned using mafft [31,32] and used to construct phylogenetic trees with fasttree2 [33].

The 16S rRNA and ITS ASVs were taxonomically classified by alignment against Greengenes (v13_8) and UNITE (v8.3) databases, respectively, using QIIME 2’s RDP naive Bayesian classifier. Thresholds: phylum (≥75% similarity, ≥90% coverage); class/order/family (≥85% similarity, ≥90% coverage); genus (≥97% similarity, ≥95% coverage); species (≥99% similarity, ≥97% coverage). Unclassified taxa at each level were labeled “unclassified”. Alpha diversity was calculated after rarefying ASV tables to 95% of the minimum sequencing depth. Differentially abundant taxa using t-tests with Benjamini-Hochberg FDR correction in STAMP. LEfSe (LDA score threshold > 3.0) identified discriminant taxa (biomarkers) between systems. Microbial co-occurrence networks (Spearman |ρ| > 0.6, *p* < 0.01) were modularized and analyzed using the igraph package (v1.2.7). Nodes were classified by topological roles: module hubs (Zi  >  2.5, Pi  <  0.62), connectors (Zi  <  2.5, Pi  >  0.62), network hubs (Zi  >  2.5, Pi  >  0.62), and peripherals (Zi  <  2.5, Pi  <  0.62) [33,34,35]. Key ecological drivers were defined as hub (module/network) and connector nodes [36]. Networks were visualized in Gephi (v0.9.2) using force-directed layouts.

## 3. Results

### 3.1. Comparative Analysis of Verticillium Wilt Suppression

Disease incidence was assessed in 13 continuous cotton monoculture (CC) and 13 maize–cotton rotation (CR) fields. The assessment revealed that maize–cotton rotation significantly reduced Verticillium wilt severity. Specifically, implementation of a short-term maize–cotton rotational system resulted in a significantly lower Verticillium wilt disease index compared to the continuous cotton monoculture system. This reduction was observed in both northern and southern Xinjiang, with an average reduction of 49.3% following short-term rotation (Figure 1).

Regional analysis showed higher disease severity in southern Xinjiang, where the average disease index in southern continuous cotton (SCC) and southern crop rotation (SCR) fields was 73.77% greater than in northern continuous cotton (NCC) and northern crop rotation (NCR) fields (Figure 1). These results demonstrate the disease-suppressive effect of rotation. Although the initial disease pressure in southern Xinjiang is greater, short-term rotation nevertheless effectively reduces the damage of Verticillium wilt in the two places. Furthermore, rotation exhibited a more pronounced inhibitory effect on the disease in southern Xinjiang, indicating greater disease suppression potential.

### 3.2. Soil Microbiome Diversity Under Short-Term Rotation

High-throughput sequencing generated 1,467,125 fungal ITS and 1,003,997 bacterial 16S rRNA quality-filtered reads, yielding 75,544 fungal and 39,789 bacterial ASVs. Contrary to the observed disease suppression patterns, α-diversity analysis revealed significant rotation effects on fungal (Shannon index: *p* = 0.00045) and bacterial communities (Shannon: *p* = 2.2 × 10^−7^) (Figure 2A,B). Regional comparisons showed significantly higher diversity in southern Xinjiang based on Chao1 and Simpson indices, but no significant regional difference in fungal Shannon diversity (*p* = 0.89). Weighted UniFrac PCoA confirmed that geography (North vs. South Xinjiang) exerted a stronger influence on β-diversity than cropping system, with bacterial communities showing greater spatial fidelity than fungal communities (Figure 2C,D). Complementing this, Principal Coordinates Analysis (PCoA) using Bray–Curtis distances further validated distinct microbial community structures associated with different cropping systems. Samples were clustered distinctly by treatment, with a clear difference between (NCR, SCR) and continuous cotton (NCC, SCC) (Figure 2C,D). This demonstrates that crop rotation not only influenced alpha diversity but also significantly altered the microbial community. These diversity patterns persisted beyond taxonomic composition, suggesting potential functional implications. Specifically, the distinct biogeographical organization of bacterial communities indicates pronounced niche specialization.

Comparative analysis revealed distinct biogeographical patterns in microbiome uniqueness. In the fungal community, the number of unique fungal ASVs in the northern crop rotation system (NCR, 1357 ± 21 SE) was 22.4% higher than that in northern continuous cotton monoculture fields (NCC, 1277 ± 18 SE, *p* = 0.034). Conversely, the number of unique fungal ASVs in southern crop rotation fields (SCR, 1268 *p* = 0.41) was lower than that in southern continuous cotton monoculture fields (SCC, 1304), indicating a region-specific response of fungal taxa to rotation management (Figure 3A). For the bacterial community, the number of unique bacterial variants in northern crop rotation fields (NCR, 26,727) was 4.7% higher than that in northern continuous cotton monoculture fields (NCC, 25,524). In contrast, bacterial diversity increased significantly in southern crop rotation fields (SCR), with 28,213 unique bacterial ASVs compared to 26,794 in southern continuous cotton monoculture fields (SCC), suggesting that bacterial communities in southern regions are more sensitive to rotation-induced disturbances (Figure 3B).

These contrasting biogeographic patterns suggest that the assembly of soil microbial communities is influenced by a combination of pedoclimatic factors and agricultural management practices. The interplay between these abiotic and anthropogenic factors appears to create region-specific ecological niches that differentially influence microbial structure, function, and interactions. Thus, the effectiveness of pathogen suppression through microbiome-mediated mechanisms may vary from region to region, depending on the local environmental context and cropping system history. This highlights the importance of considering both site-specific environmental conditions and management strategies when designing sustainable, microbiome-based disease control approaches in agroecosystems.

### 3.3. Short Rotation-Induced Shifts in Microbiome Composition and Network

Taxonomic profiling identified 9 and 12 differentially abundant fungal genera between CC and CR systems in northern and southern Xinjiang, respectively. Notably, crop rotation in northern Xinjiang was associated with distinct enrichment of *Schizothecium* (+2.8%) and *Doratomyces* (+1.9%), while depleting *Pseudogymnoascus* (−58%) and *Acremonium* (−41%) (Figure 4A). Conversely, the southern cropping system exhibited increased *V. dahliae* abundance (+3.2%) alongside a significant reduction in the *Eremomyces* (−64%), potentially indicating a loss of beneficial taxa under the southern system (Figure 4B). Strikingly, bacterial communities demonstrated stronger responsiveness to rotation than fungi, with the number of discriminant genera being substantially higher in both regions (northern: 44 genera; southern: 123 genera). Specifically, *Gemmatimonas* (+0.60%) and *Micrococcaceae* (+0.20%) dominated northern rotations, where RB41 depletion (−1.0%) was observed, suggesting alterations in soil bacterial community composition driven by plant-microbe interactions under maize–cotton rotation systems (Figure 4C). In parallel, southern systems showed *Sphingomonas* enrichment (+0.25%) concurrent with *Vicinamibacteraceae* reduction (−0.35%) (Figure 4D). Collectively, these biogeographical patterns reveal an underlying principle of microbial restructuring: bacterial communities exhibit significantly greater rotational plasticity than fungal assemblages across arid cotton ecosystems.

LEfSe analysis revealed *Hypocreales* (LDA = 4.8) and *Blastocatellia* (LDA = 5.2) as biomarkers for CC biomarkers, versus *Lasiosphaeriaceae* (LDA = 4.0) as a biomarker for CR systems (Figure 5). In the fungal community (Figure 5A,B), CC exhibited a notable enrichment of the order *Hypocreales*, whereas CR was associated with a higher abundance of the family *Lasiosphaeriaceae*. In bacterial communities (Figure 5C,D), continuous cropping was associated with a significant increase in the class *Blastocatellia*, while crop rotation correlated with a greater abundance of the order *Tepidisphaerales* (Figure 5). Furthermore, crop rotation facilitated an increased abundance of *Tepidisphaerales* and *Lasiosphaeriaceae*, suggesting a shift toward a more diverse and resilient soil microbial community structure. This restructuring likely contributes to enhanced soil resilience and disease suppression (Figure 5). The distinct fungal and bacterial biomarker profiles further provide compelling evidence of microbial community coordination in response to agricultural management, underscoring the value of integrated microbiome assessments in agroecosystem monitoring.

To characterize short-term rotation effects on soil microbiome structure, we therefore conducted a genus-level microbial co-occurrence network analysis (Figure 6). This analysis revealed rotation-induced increases in complexity in microbial co-occurrence networks. Specifically, fungal networks gained 28 nodes (+40.6%) and 146 edges, representing an 87.9% increase in edges. In contrast, bacterial networks maintained a relatively maintained node count despite a 62% increase in edges, reflecting a significant rise in inter-taxa associations and overall network density (Figure 6A,B). Notably, bacterial networks exhibited 3.2% higher connectivity than fungal counterparts (average degree 9.5 vs. 3.0), demonstrating significantly stronger rotation-driven reorganization in prokaryotic versus eukaryotic communities. Overall, topological changes in microbial symbiotic networks highlight fundamental divergences in the adaptive strategies employed by bacterial and fungal communities in response to crop rotations. Bacterial communities adapt by enhancing network connectivity, characterized by increased edge density and tighter associations between taxa. Conversely, fungal communities prioritize node expansion, reflected in the recruitment of more taxa into the network. This strategy may support greater niche occupancy and functional diversification, allowing fungi to flexibly respond to changing environmental conditions and resource inputs. These distinct responses highlight the unique ecological roles and interaction dynamics of bacteria and fungi in the soil microbiome under rotation management.

## 4. Discussion

Verticillium wilt poses a critical constraint on cotton production in Xinjiang, where the widespread adoption of long-term monoculture practices has contributed to sustain and persistence of soil-borne pathogens, particularly *V. dahliae,* not only disrupting crop yield and fiber quality but also threatening the long-term sustainability of cotton-based agroecosystems in the region. Due to the wide host range of *V. dahliae*, the strong activity of microsclerotia, and the severe shortage of disease-resistant varieties, the existing prevention and control measures are ineffective. In order to reduce the accumulation of soil-borne pathogens, cereal crops (such as wheat, rice, and corn) are often rotated with cotton [37]. Crop rotation mitigates pathogen infection through three mechanisms: inhibiting pathogen growth and reproduction or disrupting the disease cycle, enhancing the soil microecological environment, and producing direct or indirect inhibitory compounds or specific antagonistic microorganisms [20,38]. Crucially, our study demonstrates that the implementation of short-term maize–cotton rotation can significantly mitigate Verticillium wilt incidence across diverse agroecological fields in Xinjiang; by disrupting the pathogen’s lifecycle and altering soil microbial communities, rotational practices offer a viable, ecologically strategy to reduce disease prevalence and improve soil health in cropping systems (Figure 1), which is due to soil-borne diseases being closely linked to imbalances in soil microbial community structure [37], a finding that aligns with global observations of rotation-mediated pathogen suppression [39]. Notably, rotation significantly suppressed disease observed in both regions; the southern region exhibited more substantial reduction compared to northern Xinjiang (Figure 1), highlighting the geography-dependent efficacy, suggesting that region-specific edaphic factors, such as soil microbiome composition or abiotic conditions, may mediate these differential responses.

Contrary to expectations from extended rotation systems [40], short-term intervention induced significant changes in α-diversity (Figure 2A,B) alongside pronounced β-diversity reorganization (Figure 2C,D). This pattern, where functional redundancy stabilizes aggregate diversity metrics while compositional reorganization drives functional shifts, aligns with observations in arid agroecosystems. This pattern, where functional redundancy stabilizes aggregate diversity metrics while compositional reorganization promotes functional alterations, is consistent with findings made in agroecosystems that are characteristic of arid environments [41]. The significantly stronger regional influence on community structure than rotational management implicates edaphic factors (e.g., pH, organic matter) as primary microbiome determinants, consistent with findings in subtropical soils [42]. These findings support a prior study showing that rotations of one year can significantly alter rhizosphere community structure and the potential for disease resistance, particularly in arid regions that have reduced microbial baselines [39,43]. These microbial shifts in CR systems suggest rotation functions as a microbial “restructuring mechanism,” particularly effective in long-term monoculture ecosystems.

Rotation consistently enriched beneficial copiotrophs like *Sphingomonas* (Proteobacteria; +211% in SCR) while suppressing pathogens such as *Fusarium* (−64% in SCR; Figure 4B,D), mirroring maize root exudate-mediated microbiome remodeling [40,44]. Region-specific responses were evident: *Gemmatimonadetes* emergence (+182% in NCR; Figure 4C) correlates with their known adaptation to water-limited nutrient cycling, while southern shifts in *Chloroflexi*/*Acidobacteria* reflect adaptations to local pedoclimatic conditions [45]. Co-occurrence network analysis revealed fundamental differences in the structural responses of microbial communities to crop rotation. Specifically, bacterial communities demonstrated significantly greater network complexity and interconnectivity compared to fungal communities, with values that were 3.2% higher than those of fungi. In contrast, fungal networks appeared less closely connected, indicating a more modular or specialized interaction pattern (Figure 6A,B), indicating stronger rotational impacts on bacterial assembly processes. The 62% edge increase in bacterial networks versus 42% in fungi suggests rotation preferentially enhances microbial interaction complexity, a critical determinant of ecosystem resilience [4,46]. These structural shifts correlate with functional biomarkers: Hypocreales (LDA = 4.1) dominance in CC systems versus *Lasiosphaeriaceae* (LDA = 4.6) enrichment in CR (Figure 5), potentially mediating pathogen suppression through competitive exclusion. These beneficial microbes enhance network integrity in the soil microbial community, contributing to soil ecosystem stability and resilience through synergistic interactions.

Furthermore, the differential bacterial versus fungal networks showed that bacteria are more sensitive to rotation, which in turn suggests that bacteria have the potential to serve as early soil management bioindicators. Future precision agriculture should incorporate bacterial network indicators into diagnostic soil health assessments [11,47]. Regional differences in microbiome assembly (Figure 3A,B) fundamentally underscore the need for tailored soil and disease management strategies in cotton-growing regions. Notably, microbial community analysis revealed a higher degree of compositional overlap between continuous cotton (CC) and maize–cotton rotation (CR) systems in northern Xinjiang compared to southern regions (e.g., 14,987 shared bacterial ASVs) between CC and CR fields in the north, indicating a relatively stable and resilient microbial community structure despite differences in cropping practices. Such stability may contribute to the observed stronger disease suppression in northern rotations, wherein microbial network hubs increased by 37% [48]. Collectively, our finding supports the adoption of short-term (one-year) maize–cotton rotations as a practical and agronomically viable strategy for managing soil health and sustaining cotton productivity in Xinjiang’s cotton systems. This rotational scheme offers a feasible alternative that effectively disrupts pathogen persistence while maintaining land use efficiency. Mechanistically, the observed benefits of rotation appear to be mediated through targeted shifts in the soil microbiome, including the selective enrichment of stress-responsive and potentially beneficial microbial taxa, such as *Sphingomonas*, while suppressing some pathogens (e.g., *Verticillium*). This microbial restructuring leverages functional redundancy within the soil community to enhance disease suppression and overall ecosystem resilience. The strategy proved particularly effective in northern Xinjiang, where the native microbiome exhibited greater compositional stability and responsiveness to rotation, leveraging microbial functional redundancy for disease control, a strategy demonstrating particular efficacy in northern regions with higher native microbiome resilience.

## 5. Conclusions

Short-term (one-year) maize–cotton rotation significantly reduced Verticillium wilt across cotton fields in Xinjiang, with stronger suppression observed in northern regions. Rotational practice not only suppressed pathogens (e.g., *Verticillium)* but also selectively enriched beneficial microbial groups (e.g., *Sphingomonas*), which are known for their plant-protective and stress-resilient properties. Furthermore, rotation treatments significantly altered the structure of the soil microbial co-occurrence network (bacterial edge density: +62%), indicating enhanced microbial connectivity and potential functional complementarity. Compared to fungi, bacterial communities exhibited more substantial structural reorganization, highlighting their greater ecological responsiveness to cropping system changes.

Notably, cotton production systems in northern Xinjiang exhibited a dual advantage under short-term maize–cotton rotation: they not only achieved more effective suppression of Verticillium wilt but also maintained microbiome stability (core ASV conservation) across treatments, whereas southern regions showed higher baseline pathogen pressure. One-year rotations offer a viable strategy for balancing sustaining crop productivity and enhancing soil health in cotton monoculture systems. However, to fully optimize the long-term benefits of this strategy, future research should prioritize multi-year, longitudinal field trials across diverse environmental gradients. Additionally, integrating functional metagenomics and transcriptomics will be essential to unravel the specific microbial pathways and interactions responsible for disease suppression and resilience, particularly in arid agroecosystems in Xinjiang.

## Figures and Tables

**Figure 1 microorganisms-13-01968-f001:**
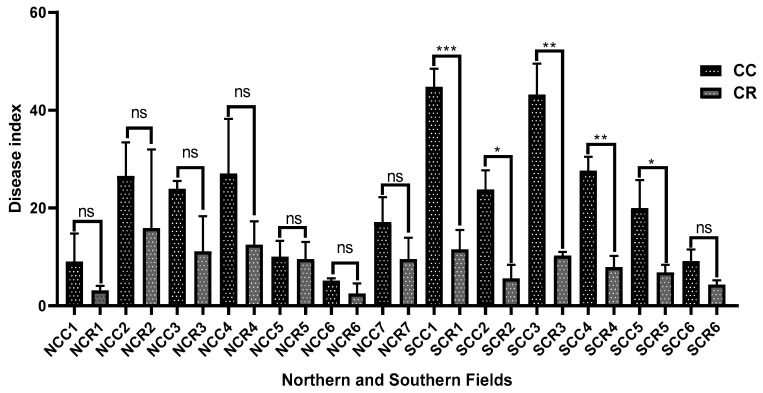
Comparison of Verticillium wilt severity between CC and CR in northern and southern Xinjiang. Two-tailed Student’s *t* test. * *p*  <  0.05; ** *p*  <  0.01; *** *p*  <  0.001; ns, not significant. The error bars represent the mean  ±  SDs.

**Figure 2 microorganisms-13-01968-f002:**
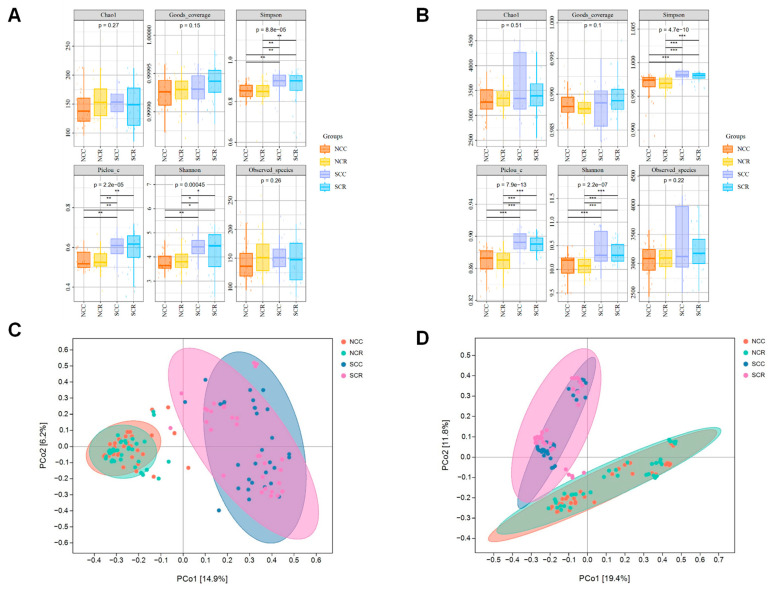
Analysis of the abundance and composition of fungal and bacterial communities in CC and CR rhizosphere soils in southern and northern Xinjiang. The α diversity of CC and CR fungal (**A**) and bacterial (**B**) communities. PCoA of fungal community (**C**) and bacterial community (**D**) based on Bray–Curtis distance between CC and CR (*n* = 130). * *p*  <  0.05; ** *p*  <  0.01; *** *p*  <  0.001; ns, not significant.

**Figure 3 microorganisms-13-01968-f003:**
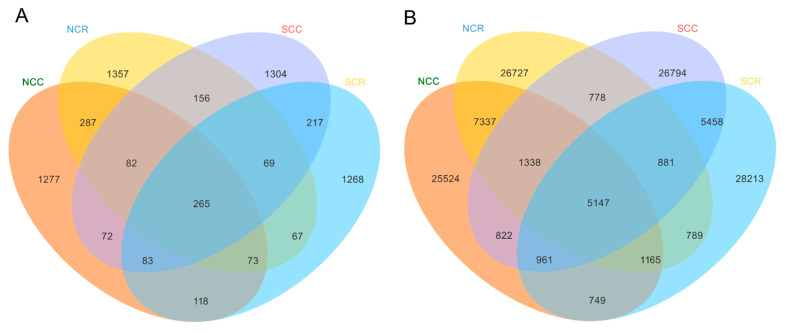
ASVs specific or common to the cotton microbiome in different cropping systems and regions. (**A**) The unique and common number of fungal ASVs among NCC, NCR, SCC, and SCR. (**B**) The unique and common number of bacterial ASVs of NCR, NCC, SCR, and SCC, and (*n* = 130).

**Figure 4 microorganisms-13-01968-f004:**
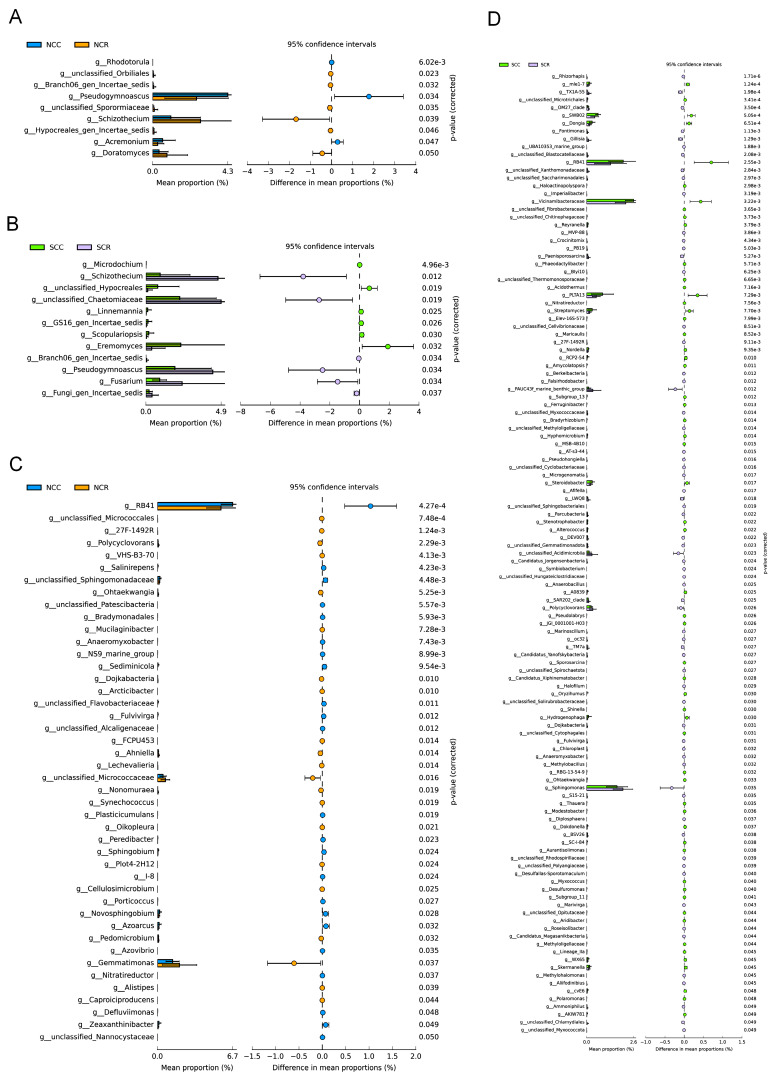
The differential abundance of CC and CR microbial genera. (**A**) Northern fungal genus (NCC vs. NCR). (**B**) Southern fungal genus (SCC vs. SCR). (**C**) Northern bacterial genus (NCC vs. NCR). (**D**) Southern bacterial genus (SCC vs. SCR) abundance comparison. The relative abundance was tested using a two-sided *t*-test, and the corrected *p*-value is shown in the figure.

**Figure 5 microorganisms-13-01968-f005:**
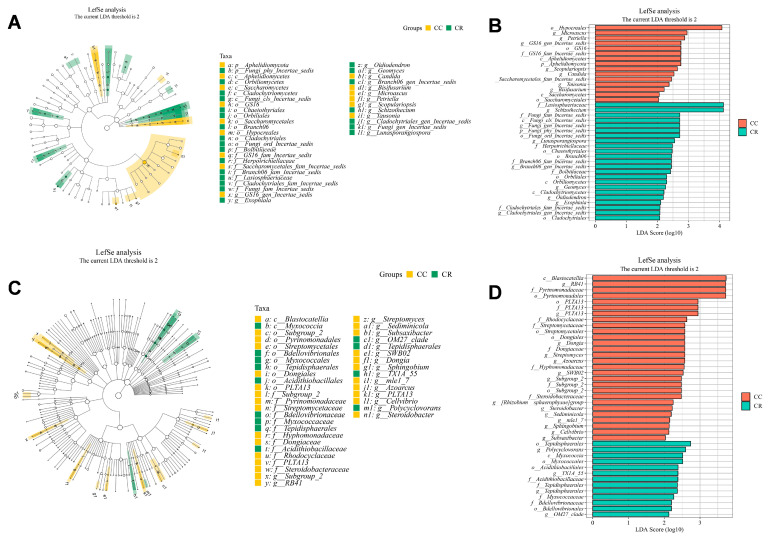
LEfSe analysis of fungal (**A**,**B**) and bacterial (**C**,**D**) community biomarkers between CC and CR. The cladogram depicted five levels of taxonomic ranks, with distinct colors representing the differential relative abundance of microbes in CC (yellow) and CR (green). Non-significant differences are denoted by hollow circles. The LEfSe bar chart highlights biomarkers that exhibit significant differences between CC and CR, with the length of the bars representing the impact of the species. The LDA score threshold was log10 (LDA score) > 2.0.

**Figure 6 microorganisms-13-01968-f006:**
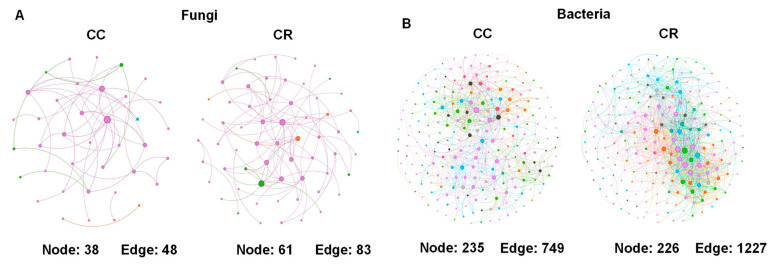
Co-occurrence network of fungal (**A**) and bacterial (**B**) communities between CC and CR based on a correlation analysis. The nodes represent unique ASVs, and the node size is proportional to its degree. Purple links represent positive correlations, while green links represent negative correlations between nodes. Orange links show mixed correlations and gray links represent neutral or weak correlations, while black links represent no correlations.

## Data Availability

Raw Illumina sequence data are deposited in the NCBI Sequence Read Archive under the accession numbers PRJNA1271755 and PRJNA1271327.

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
