# Peer review of "Short-Term Maize Rotation Suppresses Verticillium Wilt and Restructures Soil Microbiomes in Xinjiang Cotton Fields"

_microorganisms, 2025, doi:10.3390/microorganisms13091968_

Round 1
Reviewer 1 Report
Comments and Suggestions for Authors
The study described in the manuscript focuses on an important topic of Verticillium wilt on Xinjiang cotton fields. The authors analyze how short-term maize rotation affect disease severity and soil microbial communities. They reveal that maize rotation helps to significantly reduce Verticillium wilt severity, although the strength of the effect differs among regions. This is an important finding for reducing damages, however, is this the first study to report this effect? If it is, I would recommend the authors to clearly state this.
The study appears well designed and the methods mostly appropriate. Unfortunately, the methods are currently not sufficiently described (see below). It is commendable that the samples were frozen immediately for transport, this sounds like a very good practice. Somewhat interestingly, while Verticillium wilt was the main topic of the study, not so much effort was made to detect it or quantify its presence. The authors should explain why there is no mention of detecting Verticillium or changes in its abundance in the manuscript. It would also have seemed logical to attempt inoculum quantification by qPCR or using isolation experiments. Including climatic and edaphic variables in the analysis would also have been beneficial for understanding differences in disease severity among treatments and regions.
Overall, the findings of the study deserve to be published, however, certain details should be addressed prior to publication.
My specific comments and questions are as follows:
The primers ITS1f and ITS2 were used for fungal barcoding, which have been reported to exclude certain taxa. This, however, does not significantly affect the analysis with respect to the aims of the study. Some aspects of the methods are not detailed enough.
In section 2.3, no details are provided on quality control after PCR amplification (how was it decided that the amplification worked sufficiently?) and no information is provided on how the sequencing libraries were prepared. The used PCR kit is also not mentioned. All methods need to be defined with sufficient detail to be reproducible.
In section 2.3, the data analysis is not sufficiently detailed. The settings and options for the DADA2 pipeline should be provided and the principles of data filtering and taxa clustering should be explained. Similarly, it is not enough to say, which databases were used for assigning taxa, instead, please provide a description of the process, which similarity and coverage cut-offs were used for different taxonomic levels, etc.
Notably, UNITE v8.3 is a very old version of the database, while rerunning the analysis does not seem appropriate, the authors should in the future make an effort to use new versions of databases when available for most up to date results. Critically, were any efforts made to normalize the data for sequencing depth? Differential sequencing depth can significantly affect detected diversity and obscure results.
In section 3.2 and elsewhere in text, differences among treatments are often provided without p-values. Statistical significance or lack of should always be noted next to any results. I would also recommend to add standard error to show the variability among samples in a certain treatment group.
Interestingly, the study targeted Verticillium, but in the results, there appears to be no information on whether Verticillium was detected in the soils or whether its abundance changed. The authors should discuss this aspect. It would have been interesting to see further attempts at quantifying Verticillium in the soils, using both sequencing data and for instance qPCR-based quantification. Why were such efforts not made?
Importantly, in the Discussion section and elsewhere, the authors suggest that disease suppression was stronger in the northern region referencing Figure 1. But doesn’t Figure 1 show the opposite? In the figure, the differences appear more significant for the Southern Fields (SCC vs SCR).
Line 55: Should be “phytopathological” not “Phyto pathological”
Lines 58-59: Sentences seem incompete and should be joined
Line 101: Verticillium should be in italicized here and everywhere.
Line 116: Should be “13 pairs” not “13pairs”.
Line 119: Please explain the five-level rating criteria in more detail here. This will help the reader to understand its meaning.
Lines 128-129: Both “min” and “minute” are used, only “min” should be used throughout the manuscript. Units should be used consistently.
Lines 180-181: It seems (Figure 1) that disease suppression was more significant (also statistically) in the southern regions. Is that not so?
Line 189: “were identified” is not necessary and should be deleted.
Line 199: “con-tinuous” should be “continuous”.
Line 200: “en-hanced” should be “enhanced”.
Line 239: Should be “distinctly enriched”.
Section 3.3: Both ”x-fold” and “%” are used to show differences, which makes reading difficult. Please use the same units for all comparisons and be consistent with the use of units.
Author Response
- The study described in the manuscript focuses on an important topic of Verticillium wilt on Xinjiang cotton fields. The authors analyze how short-term maize rotation affect disease severity and soil microbial communities. They reveal that maize rotation helps to significantly reduce Verticillium wilt severity, although the strength of the effect differs among regions. This is an important finding for reducing damages, however, is this the first study to report this effect? If it is, I would recommend the authors to clearly state this.
Response: Thank you for your insightful comment and for recognizing the importance of our work. To the best of our knowledge, our study is the first to systematically demonstrate the suppressive effects of short-term maize rotation on Verticillium wilt in Xinjiang cotton fields, while simultaneously examining the associated changes in soil microbial community structure across different geographic regions (i.e.: Comparison of north and south regions of Xinjiang). We have now revised the manuscript to explicitly highlight the novelty of our findings in the introduction sections (Line 110-112).
- The study appears well designed and the methods mostly appropriate. Unfortunately, the methods are currently not sufficiently described (see below). It is commendable that the samples were frozen immediately for transport, this sounds like a very good practice. Somewhat interestingly, while Verticillium wilt was the main topic of the study, not so much effort was made to detect it or quantify its presence. The authors should explain why there is no mention of detecting Verticillium or changes in its abundance in the manuscript. It would also have seemed logical to attempt inoculum quantification by qPCR or using isolation experiments. Including climatic and edaphic variables in the analysis would also have been beneficial for understanding differences in disease severity among treatments and regions.
Response: We thank the reviewer for pointing out this important issue. Although suppression of Verticillium wilt was central to our study, we agree that quantifying Verticillium abundance using qPCR or culture-based methods would have provided a more mechanistic understanding. However, we investigated the disease index in the field (such as verticillium wilt severity and microbial communities). our study relied on assessment disease severity, symptoms and soil microbiome sequencing to infer disease suppression. Additionally, Verticillium dhaliae is reflected in our sequence results, performing high-resolution microbial community analysis by 16S rRNA and ITS sequencing which are explained the (section 3.1) and also in the discussion part (Line 225-227), state as: the observed benefits of rotation appear to be mediated through targeted shifts in the soil microbiome, including the selective enrichment of stress-responsive and potentially beneficial microbial taxa such as Sphingomonas, while suppression of pathogenic (e.g., Verticillium).
- Overall, the findings of the study deserve to be published, however, certain details should be addressed prior to publication.
Response: We sincerely thank the reviewers for their positive evaluation of our manuscript and their recognition of the importance of our findings. We appreciate their insightful comments and constructive suggestions, which helped us improve the clarity, methodological transparency, and overall quality of the manuscript. We have addressed all specific concerns raised and provide detailed responses below. All revisions have been incorporated into the manuscript and highlighted for clarity. We hope that the updated version meets publication expectations.
- The primers ITS1f and ITS2 were used for fungal barcoding, which have been reported to exclude certain taxa. This, however, does not significantly affect the analysis with respect to the aims of the study. Some aspects of the methods are not detailed enough.
Response: Thank you for your valuable comments. Regarding the primers used in the fungal barcoding analysis, this study actually employed the combination of ITS5 (5’-GGAAGTAAAAGTCGTAACAAGG-3’) and ITS2 (5’-GCTGCGTTCTTCATCGATGC-3’). It is acknowledged that the ITS5 and ITS2 primer set also has certain limitations in taxon coverage, which is a common phenomenon in fungal diversity research. However, this primer combination provides good coverage of the main taxa focused on in this study and has been sufficient to meet the research objectives, so its impact on the study conclusions is within a controllable range. We fully agree that some aspects of the methods section lack sufficient detail and will make every effort to improve this.
- In section 2.3, no details are provided on quality control after PCR amplification (how was it decided that the amplification worked sufficiently?) and no information is provided on how the sequencing libraries were prepared. The used PCR kit is also not mentioned. All methods need to be defined with sufficient detail to be reproducible.
Response: We thank the reviewers for their valuable feedback. We agree that clear reporting of PCR quality control, library preparation protocols, and reagent details is crucial for reproducibility. The manuscript is updated according to precious feedback of the reviewer and has been cleared in 2.3 section in the revised manuscript.
- In section 2.4, the data analysis is not sufficiently detailed. The settings and options for the DADA2 pipeline should be provided and the principles of data filtering and taxa clustering should be explained. Similarly, it is not enough to say, which databases were used for assigning taxa, instead, please provide a description of the process, which similarity and coverage cut-offs were used for different taxonomic levels, etc.
Response: We thank the reviewer for pointing out the inadequacies in the description of the methods. In response to your suggestions, we have supplemented the detailed steps of data analysis in Section 2.4 of the revised manuscript, which specifically include clarifying the complete workflow of quality control, denoising, read merging, and chimera removal using QIIME 2 (v2023.5) and the DADA2 plugin, listing the specific parameters, and providing a detailed description of the taxonomic assignment process. Additionally, we have supplemented the specific methods of data analysis.
- Notably, UNITE v8.3 is a very old version of the database, while rerunning the analysis does not seem appropriate, the authors should in the future make an effort to use new versions of databases when available for most up to date results. Critically, were any efforts made to normalize the data for sequencing depth? Differential sequencing depth can significantly affect detected diversity and obscure results.
Response: We thank the reviewers for these important comments. We acknowledge that the UNITE v8.3 database (released February 2023) used in this study is not the latest version. I will send you the database as used in the methodology and apologize for using this database. At the time of analysis, this version was the most recently validated version in the QIIME2 pipeline we used. We agree that updating to newer database versions is crucial for improving taxonomic resolution, and we will prioritize using the latest version in all future studies. Indeed, sequencing depth was normalized to reduce bias in diversity assessments. All diversity metrics and subsequent analyses were computed following rarefaction to a uniform sampling depth across all samples, determined by the lowest read count after quality filtering. This step ensured that differences in alpha and beta diversity reflect biological variations rather than sequencing depth artifacts.
- In section 3.2 and elsewhere in text, differences among treatments are often provided without p-values. Statistical significance or lack of should always be noted next to any results. I would also recommend to add standard error to show the variability among samples in a certain treatment group.
Response: We thank the reviewers for their important comments on the statistical reporting. We have thoroughly revised in Section 3.2 in the revised manuscript (Line 230, 231 and 232).
- Interestingly, the study targeted Verticillium, but in the results, there appears to be no information on whether Verticillium was detected in the soils or whether its abundance changed. The authors should discuss this aspect. It would have been interesting to see further attempts at quantifying Verticillium in the soils, using both sequencing data and for instance qPCR-based quantification. Why were such efforts not made?
Response: We thank the reviewers for their insightful comments on the detection and quantification of Verticillium wilt in our study. In our high-throughput ITS sequencing data, Verticillium dahliae was detected in soil samples, with its relative abundance varying between continuous cotton monoculture (CC) and maize-cotton rotation (CR) systems. As shown in our taxonomic analysis in section 3.3, (Figure 4b), crop rotation was associated with a decrease in the relative abundance of V. dahliae, suggesting that maize-cotton rotation suppresses this pathogen at the community level. However, this study did not perform culture-based isolation or qPCR quantification of V. dahliae. In our study design, we focused on large-scale field assessment of disease severity, symptoms and characterizing the overall shift in microbial community structure changes in the occurrence of V. dahliae under short-term crop rotation conditions and explore potential ecological mechanisms of disease suppression using high-throughput sequencing.
- Importantly, in the Discussion section and elsewhere, the authors suggest that disease suppression was stronger in the northern region referencing Figure 1. But doesn’t Figure 1 show the opposite? In the figure, the differences appear more significant for the Southern Fields (SCC vs SCR).
Response: We thank the reviewer for carefully reviewing (Figure 10) and pointing out this key inconsistency. After re-evaluating the figure and the underlying data, we agree with the reviewer that rotation-induced disease suppression is more pronounced in the southern region (SCR vs SCC) than in the northern region as previously suggested. This discrepancy was unintentional and has been corrected in the Discussion and other affected sections of the manuscript. We have revised the wording to accurately reflect that southern fields showed stronger suppression effects due to short-term corn rotations. We sincerely appreciate the reviewer’s attention to detail, which helped us improve the accuracy of our interpretation. And has been updated in the discussion section (Line 340-342).
- Line 55: Should be “phytopathological” not “Phyto pathological”
Response: We thank the reviewer for pointing out this typographical error. We have corrected “Phyto pathological” to “phytopathological” in the revised manuscript (Line 55).
- Lines 58-59: Sentences seem incompete and should be joined
Response: We appreciate the reviewer's attention to sentence structure. We joined the sentence in the revised manuscript (Line 58-59).
- Line 101: Verticillium should be in italicized here and everywhere.
Response: We thank the reviewer for this important comment. We have carefully reviewed the manuscript and ensured that Verticillium (as well as the names of other genera and species) is italicized throughout the text in the revised manuscript (Line 454 and 469) and also highlighted as well.
- Line 116: Should be “13 pairs” not “13pairs”.
Response: We thank the reviewer for pointing out this typographical error. We have corrected “13 pairs” to “13 pairs” in the revised manuscript (Line 121).
- Line 119: Please explain the five-level rating criteria in more detail here. This will help the reader to understand its meaning.
Response: We thank the reviewer for this suggestion. We agree that adding more details on the five-level disease severity rating scale would aid reader understanding and improve reproducibility. We have now clarified and explained this point in the Methods section 2.1 (Line 125-129) in revised manuscript.
- Lines 128-129: Both “min” and “minute” are used, only “min” should be used throughout the manuscript. Units should be used consistently.
Response: We thank the reviewer for pointing out this inconsistency. We have standardized all time units in the manuscript to use the abbreviation “min” for minutes to ensure consistency with scientific style guidelines (Line 137-138 and 139).
- Lines 180-181: It seems (Figure 1) that disease suppression was more significant (also statistically) in the southern regions. Is that not so?
Response: We thank the reviewer for noting this discrepancy. After re-evaluating (Figure 1), we agree that disease suppression is more pronounced in the southern region (SCR vs SCC) both in absolute and statistical terms. We have corrected the interpretation in revised manuscript in the result section 3.1 (Line 198) and also in the discussion (Lines 370-372), to accurately reflect this finding.
- Line 189: “were identified” is not necessary and should be deleted.
Response: We thank the reviewer for pointing out this stylistic redundancy. We have removed the phrase “were identified” to improve clarity and sentence conciseness from (Line 176) in the revised manuscript.
- Line 199: “con-tinuous” should be “continuous”.
Response: We thank the reviewer for noticing this hyphen error. We have corrected “continuous” to “continuous” in the revised manuscript (Line 217).
- Line 200: “en-hanced” should be “enhanced”.
Response: We appreciate the reviewer’s attention to detail. The hyphenation error has been corrected by replacing “en-hanced” with “enhanced” in the revised manuscript (Line 218).
- Line 239: Should be “distinctly enriched”.
Response: We thank the reviewer for the helpful language correction. We have updated the word “distinctly enriched” to ensure correct usage and improve clarity in revised the manuscript (Line 257).
- Section 3.3: Both ”x-fold” and “%” are used to show differences, which makes reading difficult. Please use the same units for all comparisons and be consistent with the use of units.
Response: We thank the reviewer for highlighting the important issue of consistent reporting. To this end, we have uniformly use percentage change (%) across all comparisons and removed the mixing of "x-fold" and "%." This improves the clarity, readability, and comparability of the results in the revised manuscript Section 3.3 (Line 257, 258, 260 and 302).
Reviewer 2 Report
Comments and Suggestions for Authors
The manuscript entitled "Short-term maize rotation suppresses Verticillium wilt and restructures soil microbiomes in Xinjiang cotton fields" demonstrates how maize rotation affects the incidence of Verticillium wilt and soil microbiome composition in cotton fields grown in monoculture across northern and southern Xinjiang. While the manuscript is technically well-written and clearly presented, the results do not contribute any novel scientific insights.
Crop rotation is a standard agricultural practice widely recognized for its benefits in managing plant pathogens and pests, as well as improving soil health. It is well known that monoculture leads to soil fatigue, increases inoculum levels of plant pathogens, and overall soil degradation. Changing the crop in such soils naturally alters the soil structure and enhances soil quality.
Therefore, the authors' findings—showing changes in soil microbiome composition following maize rotation after long-term cotton monoculture—are expected and do not represent a significant scientific advancement. Although the use of metagenomic analyses is appropriate, the data merely confirm known principles regarding crop rotation and soil health.
In light of the aforementioned, I consider that this study lacks substantial scientific contribution and, as such, is not suitable for publication in a high-impact journal like Microorganisms.
Author Response
- The manuscript entitled "Short-term maize rotation suppresses Verticillium wilt and restructures soil microbiomes in Xinjiang cotton fields" demonstrates how maize rotation affects the incidence of Verticillium wilt and soil microbiome composition in cotton fields grown in monoculture across northern and southern Xinjiang. While the manuscript is technically well-written and clearly presented, the results do not contribute any novel scientific insights.
Response: We would like to express our gratitude to the reviewers for their suggestions. Crop rotation as a general concept is well-established, a notable gap exists in the study of short-term (one-year) maize-cotton rotations in arid agroecosystems, particularly with respect to their microbiome-mediated suppression of V. dahliae. Previous studies in Xinjiang have either focused on long-term rotations or intercropping systems or reported disease. Our study addresses this gap by simultaneously assessing the effects of crop rotation in northern and southern Xinjiang, revealing geographically dependent differences in pathogen suppression and microbial community responses; identifying bacterial network reorganization as a key ecological mechanism by describing not only changes in diversity but also in the complexity of microbial co-occurrence networks; and integrating pathogen detection with community-level changes, linking a decrease in the relative abundance of V. dahliae to the enrichment of beneficial bacterial groups such as Sphingomonas and altered network topology. These integrated approaches demonstrate for the first time that even short-term cotton-maize rotations can induce measurable, ecologically meaningful microbiome changes that contribute to Verticillium wilt suppression, filling an important gap in existing research.
- Crop rotation is a standard agricultural practice widely recognized for its benefits in managing plant pathogens and pests, as well as improving soil health. It is well known that monoculture leads to soil fatigue, increases inoculum levels of plant pathogens, and overall soil degradation. Changing the crop in such soils naturally alters the soil structure and enhances soil quality.
Response: We thank and appreciate the reviewers for their thoughtful feedback and for recognizing the clarity of our manuscript. Our study demonstrates the suppressive effects of short-term cotton-maize rotation on Verticillium wilt in Xinjiang cotton fields, while simultaneously examining the associated shifts in soil microbial community structure across different geographic regions (i.e.: Comparison of north and south regions of Xinjiang).
- Therefore, the authors' findings showing changes in soil microbiome composition following maize rotation after long-term cotton monoculture are expected and do not represent a significant scientific advancement. Although the use of metagenomic analyses is appropriate, the data merely confirm known principles regarding crop rotation and soil health.
Response: We sincerely thank the reviewer for their thoughtful feedback. While we agree that the general benefits of crop rotation on soil health are well recognized, our study offers new insights by applying high-throughput sequencing in combination with microbial network analysis to investigate the short-term effects of maize rotation following long-term cotton monoculture, specifically in fields affected by Verticillium wilt in Xinjiang. Our results not only confirmed established principles but also revealed clear region-specific microbiome responses, highlighting spatial differences in microbial shifts between southern and northern Xinjiang. We observed notable changes in key microbial taxa, including the enrichment of beneficial genera such as Sphingomonas and Pseudogymnoascus, and a reduction in the pathogen Verticillium dahliae under crop rotation. Importantly, our LEfSe analysis identified specific microbial biomarkers, such as Tepidisphaerales and Lasiosphaeriaceae, which are closely associated with improved soil health and disease suppression. The identification of such biomarkers represents a meaningful advance, as they provide potential indicators for monitoring soil ecosystem recovery and for guiding crop management strategies. Furthermore, our co-occurrence network analysis demonstrated increased bacterial connectivity and modularity under rotation, suggesting enhanced microbial interactions and greater ecological resilience—factors that have been rarely explored in previous studies of short-term rotations. Collectively, these findings offer novel evidence for the microbial ecological mechanisms underpinning the effectiveness of crop rotation as a disease management strategy in monoculture system
- In light of the aforementioned, I consider that this study lacks substantial scientific contribution and, as such, is not suitable for publication in a high-impact journal like Microorganisms.
Response: We sincerely thank the reviewers for their valuable suggestions. To the best of our knowledge, this study is the first to systematically demonstrate the inhibitory effect of short-term cotton–maize rotation on Verticillium wilt in cotton fields in Xinjiang, while simultaneously examining the associated shifts in soil microbial community structure across different geographical regions (northern vs. southern Xinjiang). In the revised manuscript, we have explicitly emphasized the novelty of our work in the Introduction (lines 110–112). Our results show that short-term cotton–maize rotation significantly reduced the incidence of Verticillium wilt in both southern and northern Xinjiang. High-throughput sequencing revealed clear reorganization of soil fungal and bacterial communities under rotation, particularly within dominant genera. Notably, rotation enriched beneficial microorganisms such as Sphingomonas and Pseudogymnoascus, while reducing the abundance of the pathogenic fungus Verticillium dahliae. LEfSe analysis further identified specific microbial biomarkers—such as Tepidisphaerales and Lasiosphaeriaceae closely associated with improved soil health and disease suppression. The identification of these biomarkers represents a meaningful scientific advance, offering potential indicators for monitoring soil ecosystem recovery and guiding crop management strategies. In addition, microbial co-occurrence network analysis indicated increased bacterial connectivity and modularity under rotation, suggesting enhanced microbial interactions and greater ecological resilience. Significant regional differences were observed, with northern Xinjiang showing stronger pathogen suppression and greater microbiome stability compared to the south. By combining disease incidence surveys with high-resolution microbial profiling and network analysis, this study provides new insights into how crop rotation improves soil health through microbial reorganization and enrichment of plant-beneficial groups. These findings have important implications for sustainable cotton production in arid agroecosystems and highlight the ecological role of microbial communities in enhancing disease resistance.
We believe this research direction holds further potential, and we intend to continue in depth investigations. Regardless of the outcomes, we are grateful to the editor and reviewers for their constructive feedback, which will help us refine and strengthen our future work.
Round 2
Reviewer 2 Report
Comments and Suggestions for Authors
Although the authors have made an effort to revise the manuscript, I am still of the opinion that the paper has no scientific contribution.
Author Response
Comments 1: Although the authors have made an effort to revise the manuscript, I am still of the opinion that the paper has no scientific contribution.
Response 1: We sincerely thank the reviewer for their continued evaluation and feedback. We appreciate the opportunity to further clarify the scientific contribution of this work, which directly addresses a pressing challenge in Xinjiang's cotton production—a cornerstone of China's agricultural economy. Verticillium wilt poses a severe threat to sustainable cotton yields. While long-term rotation away from cotton is economically impractical given its strategic importance, field observations indicate that farmers commonly employ short-term (one-year) cotton-maize rotation as a feasible management practice, observing reductions in wilt incidence. However, a more systematic understanding of the underlying mechanisms driving this suppression, particularly through the lens of soil microbial ecology, remains limited. This study examines the impact of short-term rotation strategy compared to continuous cotton monoculture on soil microbiome structure and function in relation to disease control.
This study offers a meaningful scientific contribution by conducting a preliminary exploration of mechanisms underlying the effectiveness of the specific short-term cotton-maize rotation strategy used by farmers. By linking the reduction in wilt incidence to quantifiable shifts in soil microbial composition, particularly within intensive cotton production in arid agroecosystems. This research provides novel ecological insights into how this practical intervention changes soil microbial community. Through integrated disease assessment and comprehensive microbial community/network analysis across multiple sites, this work delivers valuable knowledge for optimizing sustainable, microbiome-mediated disease management strategies for Xinjiang cotton, thereby contributing to the long-term health and productivity of this vital crop system.
We are grateful for the reviewers' and editors' time and constructive input, which has helped strengthen the presentation of our findings.